# Predicting mechanical properties of CFRP composites using data-driven models with comparative analysis

**Ammar Alsheghri[1,2], Amna Alhammadi[3], Vassilis Drakonakis[4], Haris Doumanidis[4], Imad Barsoum[5,6]\*, Maher Maalouf[3]\***

**1** Department of Mechanical Engineering, King Fahd University of Petroleum and Minerals (KFUPM), Dhahran, Saudi Arabia, **2** Interdisciplinary Research Center for Biosystems and Machines, King Fahd University of Petroleum and Minerals (KFUPM), Dhahran, Saudi Arabia, **3** Department of Management Science and Engineering, Khalifa University, Abu Dhabi, United Arab Emirates, **4** AMDM—Advanced Materials Design & Manufacturing Limited, Nicosia, Cyprus, **5** Department Mechanical and Nuclear Engineering, Khalifa University (KU), Abu Dhabi, United Arab Emirates, **6** Department of Engineering Mechanics, Royal Institute of Technology (KTH), Stockholm, Sweden

\* maher.maalouf@ku.ac.ae (MM); imad.barsoum@ku.ac.ae (IB)

## Abstract

Carbon fiber reinforced polymer (CFRP) composites are increasingly utilized for their lightweight and superior mechanical properties. This study uses machine learning models to predict the mechanical properties of CFRP composites based on the volume fraction of carbon nanotubes (CNTs), interlayer volume fraction, glass transition temperature, and manufacturing pressure. Sixty-two samples covering nine different types of CFRPs were designed, manufactured, and experimentally tested. Three machine learning models, namely ridge regression, random forest, and support vector regression, were trained on the data and compared. The results demonstrated a high prediction accuracy for the flexural strength ($R^2 = 0.966$), flexural modulus ($R^2 = 0.871$), and the mode-II energy release rate ($R^2 = 0.903$). The study highlights the effectiveness of data-driven models in predicting key mechanical properties of CFRP composites, potentially reducing the need for extensive experimental testing and facilitating more efficient material design.

## Introduction

The increasing demand for lightweight materials with improved mechanical properties has elevated the importance of composite materials [1]. Composite materials, made of two or more materials with different mechanical properties, offer unique advantages. In general, for polymer-reinforced composites, the polymer matrix controls thermal and chemical properties such as glass transition temperature, corrosion resistance, and chemical stability [2,3], while the reinforcing material controls mechanical properties such as tensile strength and impact resistance [4–7].

Since the 1960s, carbon fibers have become essential industrial materials due to their superior mechanical properties, such as high specific strength and modulus, low thermal expansion, low density, chemical stability, and heat resistance [8]. Due to these properties, carbon fiber-reinforced polymer (CFRP) composites have been crucial in replacing metal-based

**Data availability statement:** All data used in this work is available in the Mendeley Data repository at: https://data.mendeley.com/datasets/fspdwb4mst/1

**Funding:** The author(s) received no specific funding for this work.

**Competing interests:** The authors have declared that no competing interests exist.

materials particularly in low-temperature applications such as thermal insulators [9], electronic packaging [10], biomedical implants [11], and cryogenic systems [12]. CFRPs offer a weight density 20–30% lower than conventional metals, specifically aluminum [13]. These properties have expanded their use in aerospace, aviation, sporting products, and new energy industries. For instance, composite materials account for 22% of the total weight of Airbus A380 [14].

Featherweight CFRPs are special composites formed by the extensions of carbon fibers to mimic the fractal structure of feathers [15]. The fractal structure ranges from carbon fibers (i.e., macro) to electrospun fibers (i.e., micro) and to carbon nanotubes (i.e., nano) imitating the feathers fractal architecture. These composites exploit fractal structures across dimensional scales, resulting in a feather-like high strength-to-weight ratio, particularly used in aerospace and automotive sectors [16]. CNTs-reinforced epoxy foams have also been used to improve the mechanical properties by utilizing the CNTs as enablers of porosity [17]. Reinforced foams could be selectively induced in CFRP composites and could potentially reduce their weight without significant losses in mechanical performance.

The mechanical characteristics of composite materials can be studied and predicted using a variety of approaches. Numerical cross-scale simulations, including finite element analysis (FEA) [18], offer a much quicker and less costly method compared to synthesizing, developing, and testing materials in a laboratory. Previous work used homogenization theory [19] and micromechanics [20] to predict the mechanical properties of composite materials. While the homogenization theory approximates the behavior of heterogeneous materials by treating them as if they were homogeneous, micromechanics models such as the rule of mixtures use the principles of mechanics to describe how the fibers and matrix interact to affect the overall properties of the composite. These models are rooted in physical understanding and involve detailed consideration of the composite's microstructure, including the distribution of fibers, matrix properties, fiber-matrix interactions, and load transfer mechanisms [21]. However, their accuracy can be limited by simplifying assumptions such as homogeneity of material phases [20] or the uniformity of strains (e.g., Voigt model) [22]. While they are generally based on fundamental physics, micromechanics models can struggle with highly complex or heterogeneous materials without significant adjustments or additional assumptions [23].

More recently, machine learning (ML) has become widely spread in predicting the mechanical properties of composite materials, significantly saving experimental and computational costs [6,7,24–28]. The application of ML also provided efficient solutions for the traditional difficulty associated with carrying out computational mechanics simulations involving multiscale and nonlinear behavior common in composites [6,7,26–28]. Researchers have employed various ML models such as support vector machine (SVM) [29], decision tree [29], regression models [30–32], gradient-boosted tree regression models [33], artificial neural network [34,35], regression tree [36], tree-based pipeline optimization [37], general regression neural network (GRNN) [38], Gaussian process regression [39], unsupervised bivariate clustering [40], and recurrent neural network (RNN) [41] to predict mechanical properties of composites. Pathan et al. [33] found that the time required to predict five mechanical properties using a machine learning model was about 0.5 seconds, which was 3600 times faster than using a finite element model that took about 30 minutes to make the same predictions. This excludes the time required to build and validate the FE model or the time required for data collection, preprocessing, training and validation of the ML model. Other studies used machine learning to propose optimized composite materials designs with enhanced toughness and strength [42]. Yang et al. [43] proposed using principal component analysis (PCA) and convolutional neural networks (CNN) to predict the stress-strain curve of binary composites evaluated throughout the failure path.

Zhang et al. [44] proposed a data-driven modeling approach to predict the flexural strength of continuous carbon fiber reinforced polymers (CCFRP) manufactured by fused deposition modeling (FDM). An ensemble learning model composed of eight different machine learning algorithms was adopted: multiple linear regression, SVM, the least absolute shrinkage and selection operator (LASSO), multivariate adaptive regression splines (MARS), generalized additive model (GAM), K-nearest neighbors (KNN), extremely randomized trees (Extra-Trees), and extreme gradient boosting (XGBoost). The model achieved 96.99% as a maximum value of $R^2$, predicting flexural strength with three factors: the number of carbon fiber layers, the number of concentric carbon fiber rings, and the polymer infill pattern. This study did not consider other processing factors such as pressure or the physical properties of the polymer matrix, like glass transition temperature.

Li et al. [45] trained a neural network to predict the nonlinear relationship between micro-structure and transverse mechanical properties of unidirectional CFRPs with microvoids. Their dataset consisted of artificial stochastic randomized representative volume elements (RVEs) from which they computed the transverse elastic modulus, tensile strength, and compressive strength using finite element analysis. Zhao et al. [46] used a similar approach to predict the homogenized properties of short fiber-reinforced polymer composites. The RVEs were 3D, and an ensemble machine learning model based on Extra Trees (ET), eXtreme Gradient Boosting machine (XGBoost), and Light Gradient Boosting machine (LGBM) was used. They concluded that Young's modulus of matrix, fiber, and fiber content were the most three important factors influencing the homogenized properties [46]. While ML models, especially deep learning, often encounter challenges to explain the underlying physical mechanisms, micromechanics models provide clear insights into the physical processes at the microstructural level, allowing the understanding how different factors (e.g., fiber alignment, matrix properties) affect the overall mechanical properties of the composite [47].

Previous studies illustrated the importance of data-driven models in predicting the mechanical properties of composite materials. However, based on the literature, no work has been done on the application of data-driven models to predict the experimental mechanical properties of featherweight CFRP composites with different combinations of glass transition temperatures, manufacturing pressures, the application of epoxy foam at the interlayer, the electrospinning of fibers, and the addition of CNTs. Separate studies compared relationships between two variables, such as the effect of adding CNTs in the interlayer on the flexural strength, the effect of increasing the glass transition temperature on the flexural modulus, or the effect of fiber electrospinning on the mode II energy release rate [15,17,48].

To address these gaps, this study aims to answer the following research questions:

1. How accurately can ML models predict the flexural strength, flexural modulus, tensile strength, and mode II energy release rate of CFRP composites based on the selected variables?

2. Which independent variables (CNT volume fraction, interlayer volume fraction, glass transition temperature, and manufacturing pressure) are most significant in influencing these mechanical properties?

3. Can including multiple variables improve the predictive accuracy compared to models focusing on single or limited factors?

By comparing the performance of ridge regression, random forest, and support vector regression models, this work provides a comprehensive understanding of the predictive capabilities of ML models in this context. The input variables used to build the models are experimental data composed of the volume fraction of CNTs, the interlayer volume fraction,

the glass transition temperature of the composite, and the pressure during manufacturing. Because the preparation of samples and their experimental testing are often costly, this study aims to investigate the feasibility of obtaining accurate model predictions with a low amount of data. Nine types of composites were designed, manufactured, and tested to accommodate various processing variables. The findings of this study could significantly reduce the need for extensive experimental testing, facilitating more efficient design and optimization of CFRP composites. Table 1 lists the nomenclature for all the abbreviations used in this manuscript.

## Materials and methods

The objective of this study is to predict flexural strength, flexural modulus, tensile strength, and mode II energy release rate for different types of CFRP specimens and to understand the relationship between the processing parameters of the CFRP.

### Dataset

Samples were designed and fabricated to investigate the combined effects of different processing parameters on the mechanical properties of the CFRPs [49].

### Processing parameters and properties

Four main processing parameters were varied: manufacturing pressure, carbon nanotube volume fraction, interlayer volume fraction, and glass transition temperature. These processing parameters are considered the independent variables of the data-driven models developed. To create foam at the interlayer, the epoxy resin was mixed with a hardener. All processing and manufacturing procedures, including material specifications, are described in previous work [15].

**Nine different types.** To gain a comprehensive understanding of the effects of the four parameters on the mechanical behavior of CFRP composites, this study involved nine

**Table 1. Nomenclature.**

| Abbreviations | Definition |
|---|---|
| CFRP | Carbon Fiber reinforced Polymers |
| AI | Artificial Intelligence |
| ML | Machine Learning |
| RMSE | Root Mean Square Error |
| CNT | Carbon Nanotubes |
| LR | Linear Regression |
| SVM | Support Vector Machine |
| FDM | Fused Deposition Modeling |
| MARS | Multivariate Adaptive Regression Splines |
| GAM | Generalized Additive Model |
| KNN | K-Nearest Neighbors |
| Extra-Trees | Extremely Randomized Trees |
| XGBoost | Extreme Gradient Boosting |
| GRNN | General Regression Neural Network |
| ANN | Artificial Neural Network |
| MFG | Manufacturing Pressure |
| ELSP | Electrospun |
| FE | Finite Element |

different types. The dataset included control CFRP 1 (14 samples), epoxy foam CFRP (8 samples), CNT epoxy foam CFRP (6 samples), ELSP CFRP (7 samples), CNT ELSP CFRP (7 samples), control CFRP 2 (6 samples), MFG 30 psi CFRP (4 samples), MFG 50 psi (5 samples), and MFG 70 psi (5 samples). The unequal sample sizes of some groups resulted due to the exclusions of some samples after failed manufacturing or failed mechanical testing. The total number of samples was 62. Control CFRPs 1 and 2 differ in the type of matrix polymer used, leading to substantial differences in the glass transition temperature.

**Glass transition temperature.** The temperature at which a polymer material transitions from the glassy state (brittle state) to a rubbery state (increased mobility of polymer chains) is known as the glass transition temperature, and it is below the melting point. As the polymer specimen is heated, the polymer chains start to move to a level where the polymer transitions from a glassy to a rubbery state. As the temperature increases, the polymer chains move with higher mobility until melting occurs [50]. Two different epoxy resin systems were used for the polymer matrix of the two control groups to obtain different ranges of glass transition temperatures. The details and specifications of the epoxy systems used for CFRP 1 and CFRP 2 control are detailed in previous work [15] and [16], respectively. After preparing the nine types of composites, their glass transition temperatures were measured according to the procedures described in [16]. The CFRP 1 control had an average glass transition temperature of 55 °C and the CFRP 2 control had an average glass transition temperature of 215 °C.

**Manufacturing pressure.** The mechanical and physical properties of composites can vary dramatically according to different manufacturing processes. Specifically, the properties of CFRP composites are influenced by the pressure to which they are exposed to during manufacturing. According to the literature, four different manufacturing pressures were compared: vacuum (control CFRP 2), 30 psi (MFG 30 psi CFRP), 50 psi (MFG 50 psi CFRP), and 70 psi (MFG 70 psi CFRP) [15,17,48]. Changes in the glass transition temperature of the composite are expected with variations of manufacturing pressure. Both the voids and the glass transition temperature decrease with increasing the manufacturing pressure [16]. Composite types 1 to 6 were prepared in vacuum, whereas types 7 to 9 were prepared under a manufacturing pressure in a clave (Heatcon, Seattle, U.S.A.).

**Interlayer and carbon nanotubes volume fractions.** Several techniques have been used in the literature to improve the interlayer mechanical properties of CFRPs such as the addition of epoxy foam (EPFOAM) in the interlayers or fiber electrospinning (ELSP) [15,48]. It was also revealed that the addition of carbon nanotubes (CNTs) with epoxy foam or the electrospun fibers in the interlayer assists in the consistency of multi-scale load transport at the micro-dimension and leads to an increase in tensile and flexural strengths [15]. Electrospinning produces nanofibers that highly contribute to the lamina-to-lamina ratio in a fractal structure. To produce high-quality composites, the interlayer thickness should not exceed 30 $\mu m$ [48].

In this study, four different combinations were tested, namely: EPFOAM CFRP with inter-layer volume fraction of 8%, CNT EPFOAM CFRP with interlayer volume faction of 8% and CNT volume fraction of 1.36%, ELSP CFRP with interlayer volume fraction of 6.6%, and CNT ELSP CFRP with interlayer volume fraction of 6.6% and CNT volume fraction of 1.36%. These combinations had a manufacturing pressure of 0 psi and were tested and compared with CFRP 1. The addition of CNTs with a volume fraction of 1.36% was considered based on the saturation point of the polymer during electrospinning. This percentage was decided based on trial and error. The interlayer volume fraction of EPFOAM CFRP is higher compared with ELSP CFRP because of the expansion of the foam as a result of the creation of voids. It is very important to control the interlayer volume fraction to keep the volume fraction of the fibers constant within the CFRP [15], which is approximately 55% in this study.

## Mechanical properties

The mechanical properties of composites serve as the outputs to be predicted by the data-driven models. Four mechanical properties: flexural strength, flexural modulus, tensile strength, and mode II energy release rate, were tested for the different composite groups. The specimens were tested in a screw-operated Instron machine (model 4505). Mechanical tests were performed from no loading state to failure. The following parameters were used for the flexural and mode II experiments: Data Acquisition Rate 10 point/sec, Crosshead Speed 5.0 mm/min, Temperature 23 °C, and Humidity 50%. A 10 kN load cell was used for the flexural & mode II tests (model 2518-603). The parameters for the tensile experiments were: Data Acquisition Rate 10 point/sec, Crosshead Speed 1.0 mm/min, Temperature 23 °C, and Humidity 50%. A 100 kN load cell was used for the tensile tests (model 2518-611) and a strain gauge extension-meter (model 2620-827). The Bluehill Materials Testing Software was used for acquisition and Origin software was used for processing.

**Flexural strength.** The flexural strength is the maximum stress on the outer surface of the test specimen, and it is equal to the greatest force applied before flexural failure on the cross-sectional area of sample. The samples used to measure the flexural strength were manufactured and tested according to the standard test method for the composite materials ASTM D7264 for a three-point load test on a simply supported beam that uses central loading. Overall, the test method involves supporting the bar with two supports and loading it through a nose located halfway between them [51]. A force is applied on the sample until one of the outer surfaces fails. The setup for the experiment is shown in Fig 1.

The standard span-to-thickness ratio of 32:1 was used so that failure occurs on the outer surface of the specimen only due to the bending moment. The dimensions of each experimental specimen were measured 3 times and their averages were calculated. The average thickness of the specimens was 3.58 ± 0.21 mm, the average width was 13.15 ± 0.25 mm, and the average length was 145.98 ± 0.12 mm, which was around 20% longer than the support span. The stress at any point was calculated by the following equation:

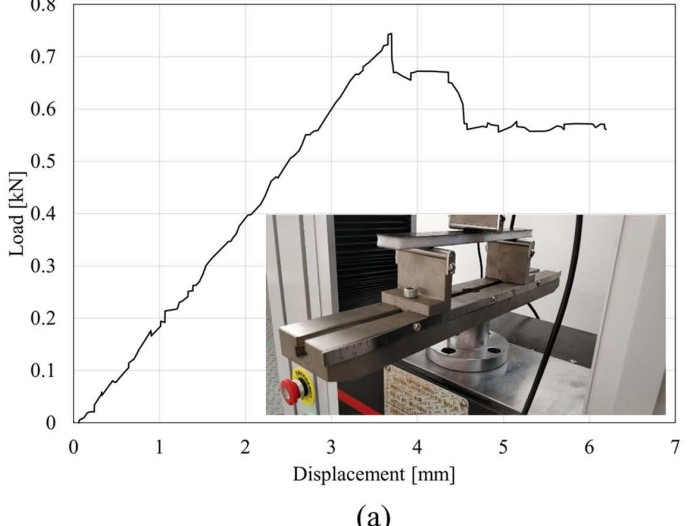

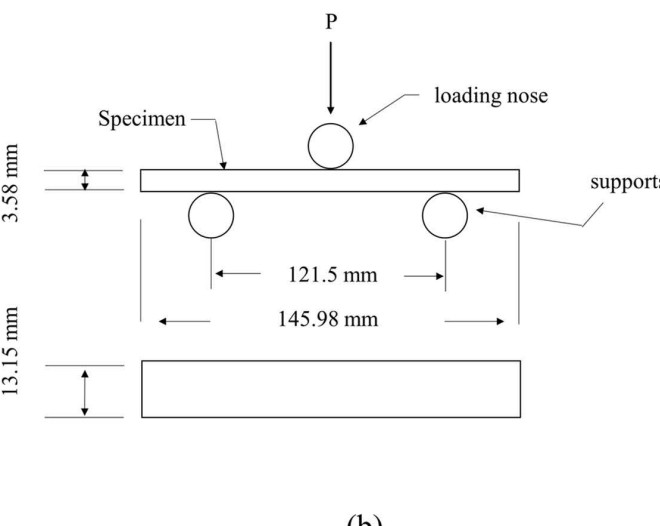

(a)                                                                 (b)

**Fig 1. (a) Representative flexural load-displacement curve; (b) schematic of three-point loading test (ASTM D7264).**

$$\sigma = \frac{3PL}{2bh^2} \tag{1}$$

where $\sigma$ is the stress at the outer surface [MPa], $P$ is the applied force [N], $L$ is the support span [mm], $b$ is the beam width [mm], and $h$ is the beam thickness [mm]. The flexural strength represents the maximum stress on the outer surface of the test specimen and is calculated based on the maximum force applied before failure. Fig 1 shows a flexural load-displacement curve that has been plotted by testing one sample of CFRP, and illustrates how the CFRP behaves before and after reaching the maximum stress.

**Flexural modulus.** The flexural modulus is the ratio between the stress and strain ranges for a loaded specimen that is tested in flexural mode within the elastic region. It can be calculated by measuring the slope of the initial linear part of the stress-strain curve using the following equation [50]:

$$E_f = \frac{\Delta\sigma}{\Delta\varepsilon} \tag{2}$$

where $E_f$ is the flexural modulus [MPa], $\Delta\sigma$ is the difference in flexural stress between the two selected strain points [MPa], and $\Delta\varepsilon$ is the difference between the two selected strain points. The flexural modulus value was obtained at 0.2% of strain.

**Tensile strength.** Using the standard test method for composite materials ASTM D3039 [52], tensile properties were calculated along the direction of loading. The samples were limited to fiber reinforced composites with balanced laminates and symmetric test directions. Specimens were also tested under tensile testing. The crosshead of the machine in the tensile test had a speed of 1 mm/min and the test was performed from no loading state to failure. Under the grips of a mechanical testing machine, a thin flat strip of material with a fixed rectangular cross-section was monotonically loaded in tension while the force was recorded. The greatest force carried before failure was used to determine the ultimate strength of the material upon division with the initial cross-sectional area of the specimen. The stress-strain response of the material was established by measuring the coupon strain with extensometer sensors, from which the ultimate tensile strain was calculated. The ultimate tensile strength was calculated as [52]:

$$S_{ut} = \frac{P_{max}}{A} \tag{3}$$

where $S_{ut}$ is the ultimate tensile strength [MPa], $P_{max}$ is the maximum force before failure [N], and $A$ is the initial cross-sectional area [mm$^2$]. The dimensions of each unidirectional fiber specimen were measured 3 times and their averages were calculated. On average, the specimens were $2.23 \pm 0.13$ mm thick, $27.55 \pm 0.43$ mm wide, and $253.0 \pm 1.00$ mm long. The tensile test setup with a representative tensile load-displacement curve is shown in Fig 2.

**Mode II energy-release rate.** The mode II interlaminar fracture-toughness was determined through three-point bending test of End Notch Flexure (ENF) specimens [48,53–55]. Like the tensile test, the composite material forms for this test were also limited to fiber-reinforced composites with balanced laminate and symmetric test directions. The mode II energy-release rate $G_{IIC}$ was calculated from the elastic beam theory as [48]:

$$G_{IIC} = \frac{9a^2 P_{max}^2 C}{2\mathrm{w}\left(3a^3 + 2L^3\right)} \tag{4}$$

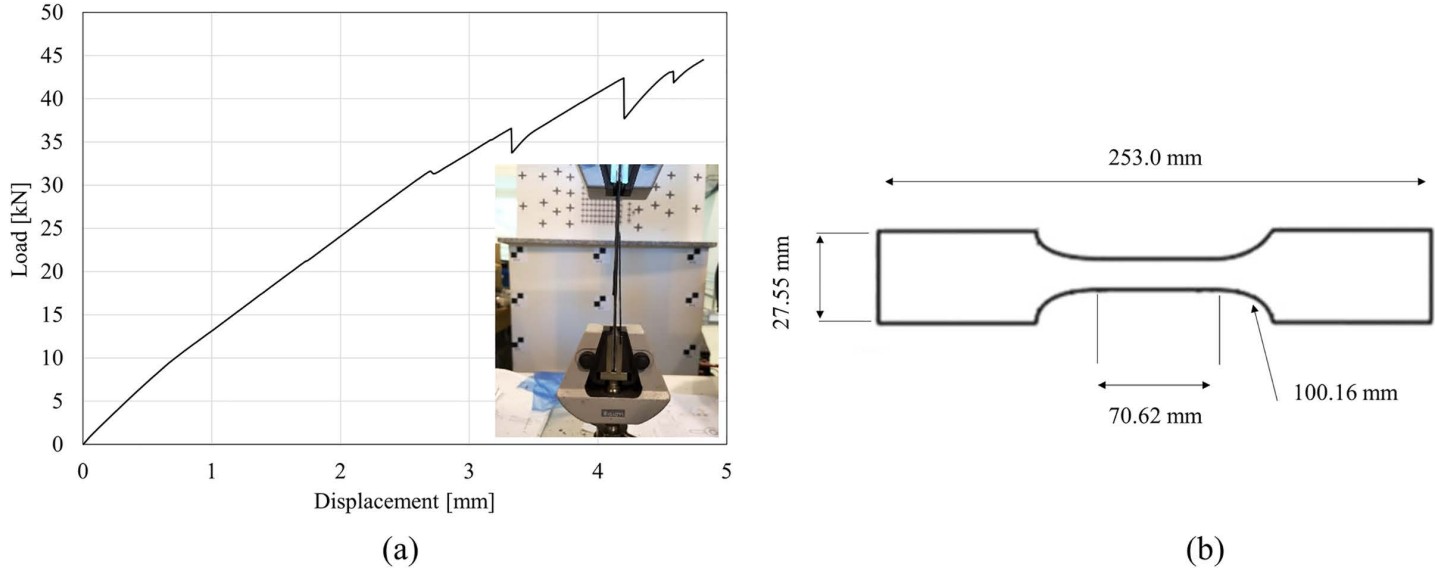

**Fig 2. (a) Tensile test set-up with a representative load-displacement curve; (b) schematic of standard tensile test method for composite materials (ASTM D3039).**

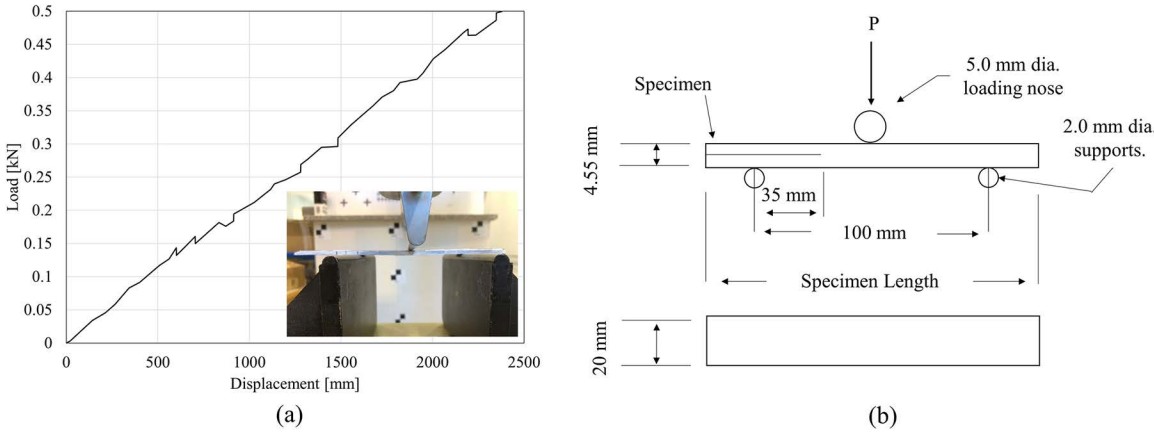

**Fig 3. (a) A representative load-displacement curve obtained from mode II fracture mechanics testing; (b) schematic of end notch flexure specimen.**

where $a$ is the crack length [m], $P_{max}$ is the maximum load [N], $C$ is the sample compliance [$Pa^{-1}$], $w$ is the sample width [m], and $L$ is the distance between central load and support [m]. Higher values of energy release rate indicate interlayer improvement. To achieve crack growth stability, the initial crack length was chosen to be equal to 70% of the half unsupported length of the specimen [54,55]. The mode II fracture mechanics testing is shown in Fig 3 with a representative load-displacement curve.

## Machine learning methods

Three machine learning models were carefully selected to cater for a small amount of training data: ridge regression, random forest regression, and support vector regression. These models provide a high degree of model generalization, which is highly important for limited data. As

with any statistical test, regression analysis requires the satisfaction of several assumptions, which have been checked and are described in the supporting information. In addition, the model refining was checked, and the variable validity was reviewed.

**Ridge regression.** Ridge regression was chosen due to its ability to handle multicollinearity among independent variables, often present in materials science data. By including a penalty term (L2 regularization), ridge regression reduces the variance of the model without significantly increasing the bias, making it well-suited for datasets with a limited number of samples. This model is particularly effective when there are many correlated predictors, as it helps in maintaining model simplicity and avoiding overfitting [56].

For each dependent variable, the linear regression model takes the following form:

$$y = X\beta + \varepsilon \tag{5}$$

where $X \in R^{N \times d}$ is the independent variable, $y \in R^N$ is the dependent variable, $\beta$ is effectively a quantification of the effect of the independent variable on the dependent variable, and $\varepsilon \in R^N$ is the error which also represents an adjustment to find the best fit for the model. The objective is to minimize the sum of squared error given by:

$$SSE = (y - X\beta)^T (y - X\beta) \tag{6}$$

This can be achieved by the following equation [57]:

$$\hat{\beta} = (X^T X)^{-1} X^T y \tag{7}$$

Ridge regression shrinks the regression coefficients by a penalty term called the L2-norm (square regulation), where L2-norm is the sum of the squared coefficients. The sum of squared error for the ridge regression is represented as:

$$SSE = \frac{1}{2}(y - X\beta)^T (y - X\beta) + \frac{\lambda}{2} \| \beta^T \beta \|^2 \tag{8}$$

where lambda ($\lambda$) is the value of the penalty, as $\lambda$ increases, the effect of the shrinkage penalty increases, and the ridge regression term will go toward zero [58,59]. A value of 0.0001 was used for $\lambda$. The solution to the previous equation is given as:

$$\hat{\beta} = (X^T X + \lambda I_d)^{-1} X^T y \tag{9}$$

where $I_d$ is the identity matrix. Compared to ordinary linear regression, ridge regression mitigates multicollinearity by accepting a small amount of bias to reduce variance and the error, thus helping to improve prediction accuracy and improve model generalization, particularly when there is a small amount of training data is available [57].

**Random forest.** Random forest regression uses an ensemble technique based on the use of many regression decision trees, which have numerical values in their leaves, providing a stable prediction with less sensitivity to changes in the input data. Bootstrap samples of the training data and random feature selection are used to build decision trees, also known as estimators. The estimators' predictions are combined (by majority vote or average) to make a final prediction by the random forest [60]. In this work, we used 100 regression decision trees as estimators for our random forest model. Random forest was chosen for its robustness and capability to model complex nonlinear relationships. Additionally, random forests provide

feature importance metrics, which are valuable for understanding the influence of different processing parameters on the mechanical properties of CFRPs.

**Support vector regression (SVR).** Support vector regression stems from the support vector machine technique, which is less sensitive to the dimensionality of the input [61]. SVR was selected for its ability to model nonlinear relationships through the use of kernel functions. It is less sensitive to outliers and can handle a small number of data points effectively. In support vector machines, a hyperplane is found in an N dimensional space to provide a best-fitting margin to classify data; N is also the number of features and number of independent variables. Compared with simple linear regression that uses ordinary least squares, support vector regression uses an $\varepsilon$ insensitive region, allowing for a margin of acceptable error which acts as a buffer for the machine learning model.

$$f(x) = \{w, x\} + b \qquad (10)$$

$$minimize: \ \frac{1}{2}\|w\|^2 + C\sum_{m}^{i=1}\left(\xi_i + \xi_i^*\right) \qquad (11)$$

$$\text{Such that:} \ \begin{cases} y_i - f(x_i) \le \varepsilon + \xi_i^* \\ f(x_i) - y_i \le \varepsilon + \xi_i \\ \xi_i, \ \xi_i^* \ge 0, \ i = 1, \ 2, \ 3,\ldots,m \end{cases} \qquad (12)$$

where $w$ is the weight vector, and $b$ is the bias, $C$ is a regularization parameter that adds a penalty for each misclassified data point, $\xi_i$ and $\xi_i^*$ are slack variable and known as the support vectors because they support the formation of the structure of the buffer region. The error is a measure between the slack variables and the margins of the insensitive region. The objective of the SVR is to find the best hyperplane to fit the data within a threshold value $\varepsilon$. During training, the model ignores training samples whose prediction is close to their target, helping generalization.

In this study, a grid search was used for each iteration to find the best combination of hyperparameters. The best tuning of these hyperparameters is totally dependent on the nature of the data used. The following values were used for the grid search: $C = $ [0.1, 1, 10, 100] and kernel = [rbf, poly, and sigmoid]. When using a Gaussian rbf kernel, another parameter $\gamma$ is needed to control the curvature of the kernel. A high value of $\gamma$ corresponds to high curvature and low value for $\gamma$ corresponds to low curvature. The grid search considered $\gamma = $ [1, 0.1, 0.01, 0.001].

## Training and evaluation

Fig 4 illustrates the methods and steps adopted in this study including data preprocessing and training procedures.

**Date pre-processing.** The sample size for the flexural strength and flexural modulus with their corresponding independent variables was 62, while for the remaining two dependent variables (tensile strength and mode II energy-release rate), the sample size was 42. After experimental data was collected, the type of composite was represented by an independent dummy variable because of its discrete values. A one-hot encoder for the types was created. The z-score function was used to normalize the independent variables (without type) by subtracting the mean from the values and dividing them by the standard deviation. After that, the data was divided into input-output to feed the regression models.

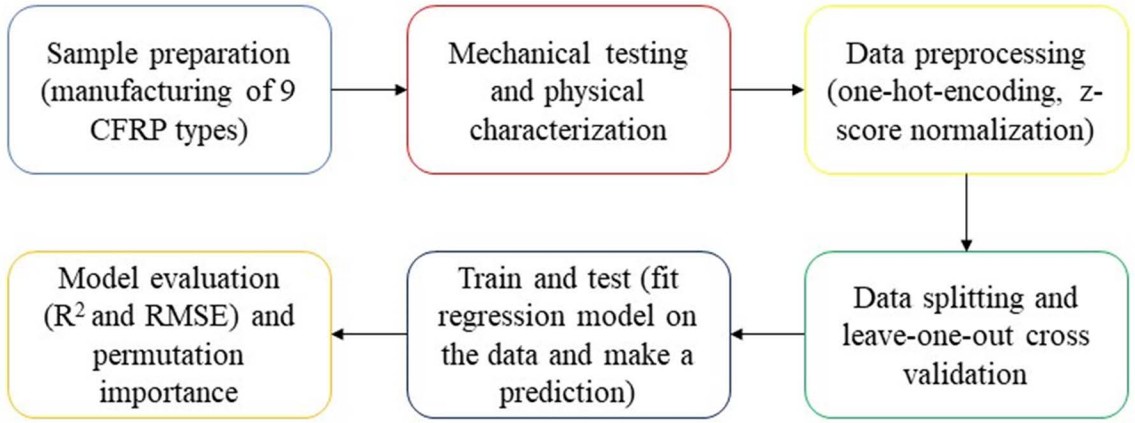

**Fig 4. Methodology chart illustrating steps of data preparation and model training.**

**Model training and validation.** The leave-one-out (LOOCV) method was used during training where the model is trained on all the data except one for N-1 times, and a prediction is made for that point. The RMSE and R² are evaluated for each trained model and an average of (N-1) is calculated for RMSE and R². A feature permutation importance was also conducted to find the most important features for each model. An average of (N-1) was considered for the overall feature important for each model.

Training and prediction were conducted using a Dell laptop with Intel(R) Core(TM) i5-7200U CPU @ 2.50GHz with 8 GB of installed RAM. For the flexural modulus model, training and prediction lasted 8.35 sec for random forest, 8.56 sec for support vector regression, and 80.51 sec for support vector regression. Pandas, scipy, and scikit-learn python packages were used to process the data and build the ML models.

**Performance metrics.** The root mean square error ( $RMSE$ ) and coefficient of determination ( $R^2$ ) have been widely used to evaluate the performance of ML regression models [39,57,62]. Assuming that $\check{y}_i$ is the predicted value of the i-th sample and $y_i$ is the true value, $RMSE$ corresponds to the expected value of the error estimated over $n$ samples in the same units of the variable of interest. $R^2$ represents the proportion of variance of $y$ that can be explained by the independent variables. The best scores for $RMSE$ and $R^2$ are zero and one, respectively.

$$RMSE\left(y, \hat{y}\right) = \sqrt{\frac{1}{n_{samples}} \sum_{n-1}^{i=0} \left(y_i - \hat{y}_i\right)^2} \tag{13}$$

$$R^2\left(y, \hat{y}\right) = 1 - \frac{\sum_{i=1}^{n} \left(y_i - \hat{y}_i\right)^2}{\sum_{i=1}^{n} \left(y_i - \overline{y}\right)^2} \tag{14}$$

## Results

Table 2 presents the mean and standard deviation values of nine different CFRP composite types. Comparing control 1 and control 2 shows that as the glass transition temperature of the composite increases, the flexural strength and flexural modulus increase. Nevertheless, higher manufacturing pressures result in higher mechanical properties, but relatively lower glass

transition temperatures. The addition of carbon nanotubes (CNTs) and electrospinning play a significant role in toughening the composites. Adding epoxy foam at the interlayer with CNTs increases flexural strength, flexural modulus, and mode II energy-release rate, but does not appear to have a significant effect on tensile strength. While it seems feasible to observe general trends, it is challenging to derive physics-based analytical models to quantify the effect of the different process parameters on the mechanical properties because of the highly complex and nonlinear relationships involved.

Table 3 shows the corresponding accuracies measured by the coefficient of determination $R^2$ and the root mean squared error RMSE for each mechanical property as a dependent variable obtained using the three developed machine learning models based on LOOCV. Flexural strength, flexural modulus, and mode II energy-release rate demonstrated a high $R^2$ value and a low RMSE value for the three models. However, the tensile strength had a low $R^2$ and a high RMSE value for all the models. Table 3 lists the results of the models trained without considering Type as an independent variable. The table shows that the highest accuracy for flexural strength and flexural modulus was obtained with the ridge regression model (the other two models were very close). The support vector regression model demonstrated the highest accuracy for tensile strength, whereas the random forest model showed the highest accuracy for mode II energy-release rate. It was also observed that normalizing the output dependent variables did not have any effect on the obtained results.

For each dependent variable, a feature permutation analysis was performed to identify the most important independent variables using the model with the best accuracy. For flexural strength, flexural modulus, and mode II energy-release rate, the actual versus the predicted values and feature permutations are presented for the model with the highest $R^2$ in Figs 5–7,

**Table 2. Mean and standard deviation values of glass transition temperature and mechanical properties nine different types of CFRP composites.**

|  | Type | Glass Transition Temperature (°C) | Flexural Strength (MPa) | Flexural Modulus (MPa) | Tensile Strength (MPa) | Mode II energy release rate (kJ/m²) |
|---|---|---|---|---|---|---|
| Control 1 | 1 | 55.16 ± 1.19 | 201.61 ± 13.11 | 25.71 ± 1.49 | 237.10 ± 3.95 | 331.83 ± 33.62 |
| Epoxy Foam | 2 | 57.21 ± 2.93 | 281.75 ± 16.52 | 40.14 ± 4.98 | 238.86 ± 5.85 | 777.80 ± 105.1 |
| CNT Epoxy Foam | 3 | 53.06 ± 2.45 | 316.21 ± 31.62 | 39.71 ± 3.04 | 245.16 ± 11.94 | 1557.44 ± 402.6 |
| ELSP | 4 | 54.29 ± 1.04 | 266.56 ± 21.59 | 35.86 ± 3.29 | 243.76 ± 3.26 | 654.81 ± 33.19 |
| CNT ELSP | 5 | 54.04 ± 0.42 | 285.33 ± 12.55 | 37.00 ± 2.16 | 247.74 ± 5.41 | 847.74 ± 68.44 |
| Control 2 | 9 | 226.83 ± 5.67 | 448.69 ± 40.38 | 41.83 ± 6.08 | – | – |
| Pressure–30 | 6 | 217.75 ± 4.86 | 522.44 ± 42.32 | 47.00 ± 4.69 | – | – |
| Pressure–50 | 7 | 211.80 ± 3.49 | 548.80 ± 5.40 | 51.10 ± 1.34 | – | – |
| Pressure–70 | 8 | 204.80 ± 5.26 | 607.05 ± 17.82 | 58.70 ± 0.84 | – | – |

**Table 3. Summary of the RMSE and $R^2$ from ridge regression, support vector regression, and random forest based on LOOCV without including the Type as an independent variable. Note: only independent variables were normalized. Asterisks correspond to the highest obtained values with respect to the three models.**

|  | Ridge Regression | | Random Forest | | SVM | |
|---|---|---|---|---|---|---|
|  | RMSE | $R^2$ | RMSE | $R^2$ | RMSE | $R^2$ |
| Flexural Strength | 25.392 | 0.966* | 27.23 | 0.960 | 30.751 | 0.949 |
| Flexural Modulus | 3.5799 | 0.871* | 3.803 | 0.854 | 3.859 | 0.850 |
| Tensile Strength | 7.015 | 0.0712 | 7.172 | 0.0288 | 6.352 | 0.238* |
| Mode II Energy-Release Rate | 259.519 | 0.657 | 235.98 | 0.716* | 309.27 | 0.512 |

*refers to the model with the highest $R^2$ value.

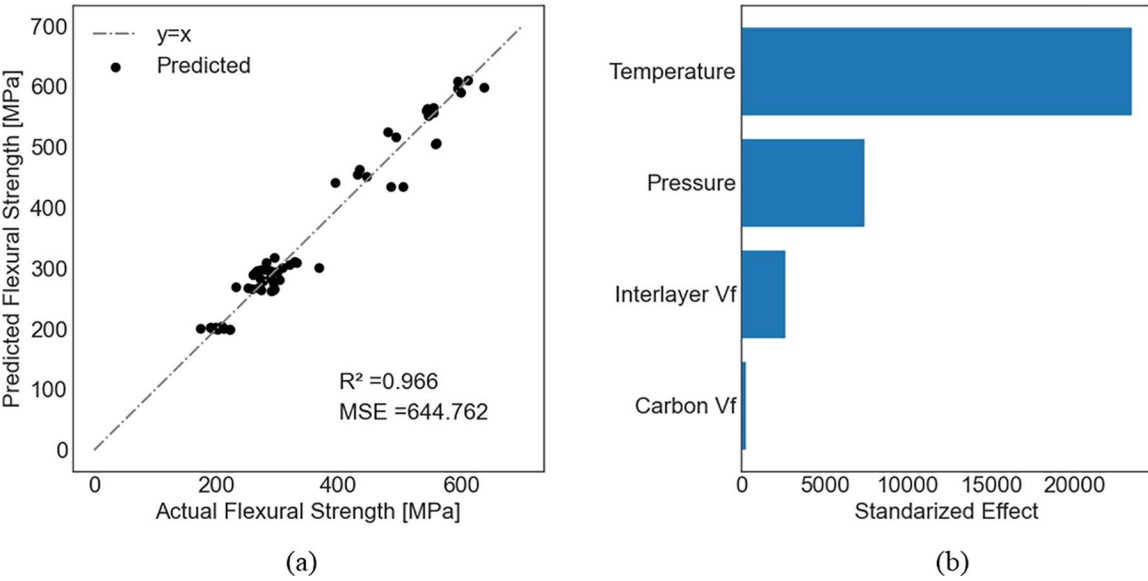

(a)                                                                                              (b)

**Fig 5. Prediction of flexural strength using ridge regression.** (a) Predicted versus actual values; (b) Feature premutation.

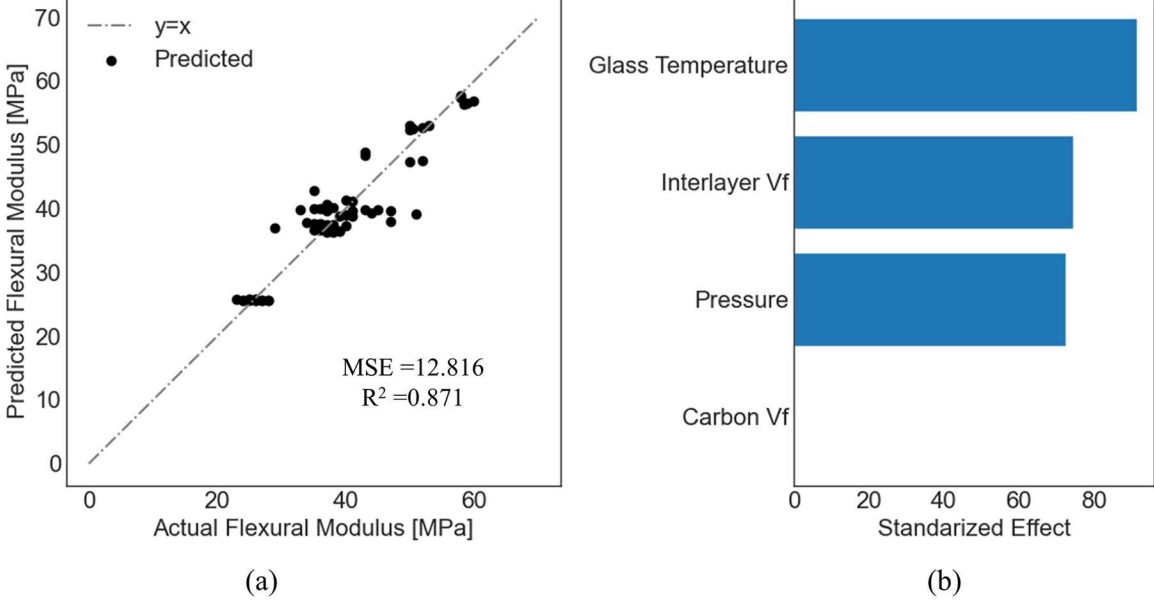

(a)                                                                                              (b)

**Fig 6. Prediction of flexural modulus using ridge regression.** (a) Predicted versus actual values; (b) Feature premutation.

respectively. Comparisons between experimentally measured and predicted properties help to assess the effectiveness and suitability of the established models to predict the mechanical properties of composites. A point closer to the diagonal line indicates that the predicted value equals the actual value.

It is noted that the glass transition temperature was the most important for both the flexural strength and flexural modulus, while the carbon volume fraction was the least important. For flexural strength, the pressure was more important than the interlayer volume fraction.

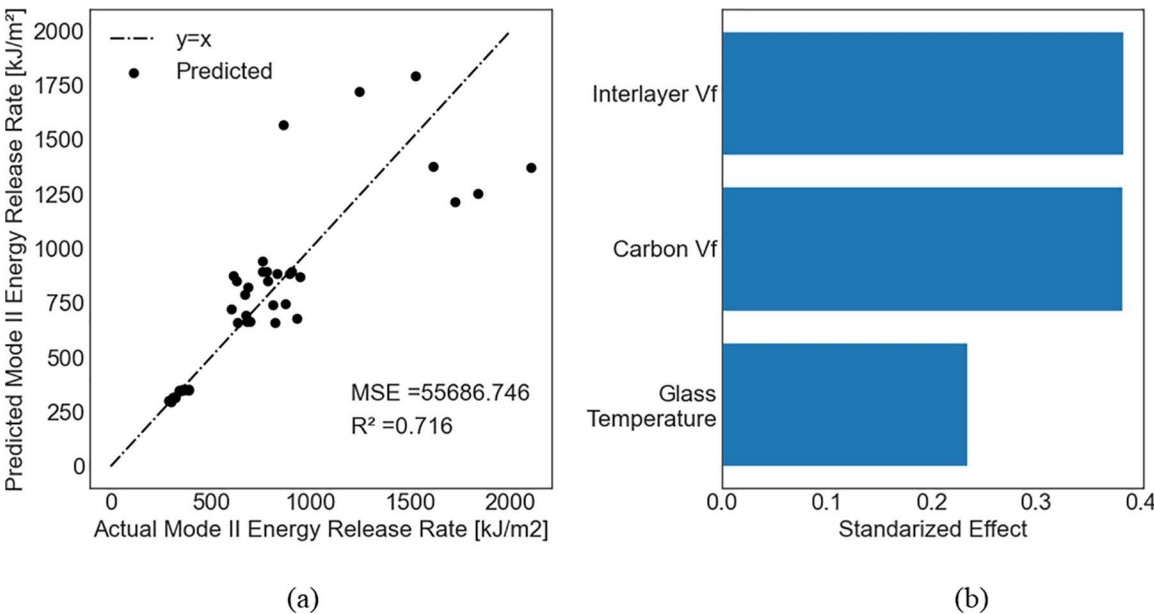

**Fig 7. Prediction of mode II energy-release rate using the random forest model without including the Type variable.** (a) Predicted versus actual values; (b) Feature premutation.

On the contrary, the interlayer volume fraction was more important than the pressure for the flexural modulus.

When considering the flexural modulus of the composite, it is observed that more independent variables have significant contributions compared with flexural strength, except for the CNT volume fraction, which had almost no effect on the modulus. Fig 7 shows that the interlayer volume fraction is the most important independent variable for the mode II energy-release rate based on the RF model followed by the volume fraction of the CNTs and the glass transition temperature. However, the contributions of both the CNT volume fraction and the interlayer volume fraction were very close. It is observed that as the actual values of the mode II energy-release rate increase, the deviations from the predicted values also increase.

Table 4 shows the predictions of the models after including Type as an independent variable. The Type feature entails intrinsic properties that cannot be explicitly measured or that are difficult to accurately measure, such as the fractal dimension of the composite [63] and CNTs geometric characteristics [64]. Table 4 indicates changes in predicted values after the inclusion of Type compared with Table 3, which did not include the Type. The comparison between both tables reveals that the inclusion of Type did not have a substantial change in flexural strength or flexural modulus. In contrast, the prediction accuracy of the tensile strength decreased, and the prediction accuracy of the mode II energy-release rate increased. Furthermore, according to Table 4, the random forest model scored the highest accuracy for flexural strength ($R^2 = 0.964$), flexural modulus ($R^2 = 0.859$), and tensile strength ($R^2 = 0.036$), while the ridge regression had the best accuracy for mode II energy release rate ($R^2 = 0.903$).

For feature permutation, no significant changes occurred for flexural strength, flexural modulus, or tensile strength. However, Fig 8b shows that the Type variable is more important than both the CNT volume fraction and the glass transition temperature. It also reveals that mixing CNTs with epoxy foam in the interlayer had the greatest influence on increasing the mode II energy release rate, followed by electrospun CFRP, epoxy foam CFRP without CNTs, electrospun CFRPs with CNTs, and control 1.

**Table 4. Summary of the RMSE and R² from ridge regression, SVM, and RF based on LOOCV with including the Type as independent variable. Note: only independent variables were normalized. Arrows are used to indicated changes in the values compared with results obtained in Table 3.**

| | Ridge Regression | | Random Forest | | SVM | |
|---|---|---|---|---|---|---|
| | RMSE | R² | RMSE | R² | RMSE | R² |
| Flexural Strength | 26.213 ↑ | 0.963 ↓ | 26.118 ↓ | 0.964↑* | 31.431 ↑ | 0.947 ↓ |
| Flexural Modulus | 3.898 ↑ | 0.847 ↓ | 3.737 ↓ | 0.859↑* | 4.399 ↑ | 0.805 ↓ |
| Tensile Strength | 7.251 ↑ | 0.008 ↑ | 7.147 ↓ | 0.036↑* | 7.455 ↑ | -0.049 ↓ |
| Mode II Energy-Release Rate | 137.916 ↓ | 0.903↑* | 180.945 ↓ | 0.833 ↑ | 258.947 ↓ | 0.658 ↑ |

*refers to the model with the highest R² value.

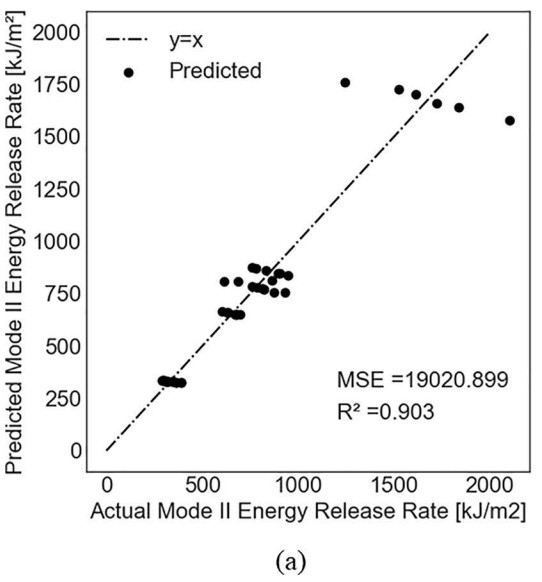 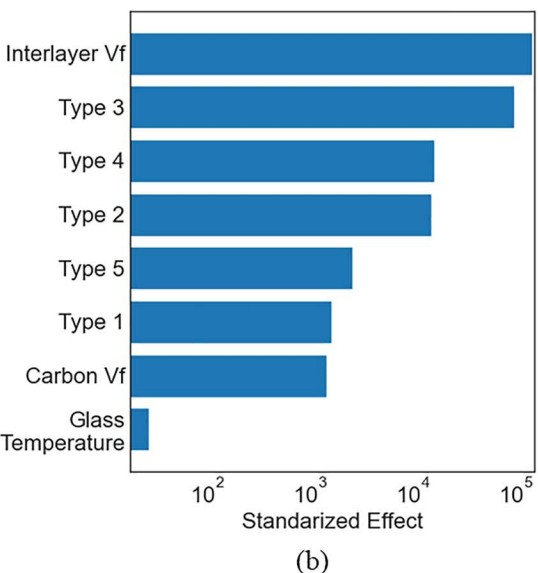

(a)                                             (b)

**Fig 8. Prediction of mode II Energy Release Rate using ridge regression with including the type.** (a) Predicted versus actual values; (b) Feature premutation.

Among the important contributions of the current study is the inclusion of Type as a dummy independent variable, which revealed a significant increase in the accuracy for predicting mode II energy release rate (about 25% increase with the ridge regression model). This conclusion was different for the flexural strength or modulus. For example, the ridge regression analysis was repeated considering the Type as a dummy variable and the change in R² was about 0.3% only in the case of flexural strength and 2.4% in the case of flexural modulus (Tables 3 and 4). Based on that and after checking the normality of errors, homoscedasticity, and the independence of errors, two simple linear formulas were proposed to calculate the flexural strength [MPa], $S_{strength}$, and flexural modulus [MPa], $E_f$, for CFRP composites as functions of CNTs volume fraction, $V_{f,carbon}$, interlayer volume fraction, $V_{f,interlayer}$, glass transition temperature [°C], $T_{glass}$, and manufacturing pressure [psi], $P_{MFG}$, based on multilinear regression as follows (see details in supporting information):

$$S_{strength} = -6.32 \times 10^{11} + 0.0864\, V_{f,carbon} + 0.2688\, V_{f,interlayer} + 0.7905\, T_{glass} + 0.4436\, P_{MFG} \quad (15)$$

$$E_f = -8.735 \times 10^{-11} + 0.03987\, V_{f,carbon} + 0.61138\, V_{f,interlayer} + 0.67250\, T_{glass} + 0.59851 P_{MFG} \quad (16)$$

## Discussion

Although typical studies on modeling the mechanical properties of CFRPs focus on predicting flexural strength [30,44], this study attempts to investigate the prediction of the flexural modulus, tensile strength, and mode II energy-release rate in addition to the flexural strength of CFRP composites. Most of the types of CFRPs used in this study are classified under featherweight composites because they contained a combination of CNTs with epoxy foam or with electrospinning of fibers. The intrinsic complexity of the featherweight CFRP composites is a product of the material involved (i.e., different matrix polymers, fibers, CNTs, etc …), architecture (i.e., layouts, fractal dimension, epoxy foam interlayer, CNTs geometric properties such as external radius and number of walls [64]), and processing (i.e., electrospinning of fibers, application of manufacturing pressure, etc). There is no doubt that no physics-based analytical model could predict the mechanical properties of CFRP composites based on the variables mentioned above due to the high nonlinearity involved. Therefore, this study suggests using robust machine learning models with a focus on the explainability of these models using feature permutation.

The independent variables considered in this work were carefully selected based on previous work [15,17,48,64,65] and included the volume fraction of CNTs, interlayer volume fraction, glass transition temperature, and the pressure during manufacturing. Nine types of CFRP composites were produced to design the experiments. Linear and non-linear machine learning models were selected to investigate the effect of using different types of composites and processing parameters. The mechanical properties measured experimentally and listed in Table 2 agree well with the published literature [15,17,44,48]. The developed data-driven models help in predicting key mechanical properties of CFRP composites with new combinations of processing parameters, potentially reducing the need for extensive experimental testing and facilitating more efficient material design. For example, the trained ridge regression model predicted higher values for the flexural strength (688.14 MPa) and flexural modulus (68.31 MPa) with a combination of an interlayer volume fraction of 0.066, a CNT volume fraction of 0.0136, a manufacturing pressure of 70 psi, and a glass transition temperature of 204.8 C.

Our experimental work also revealed no significant change in the tensile modulus during the tensile testing because the fibers were dominant in the tensile load transfer. Therefore, the tensile modulus results were not included. The average tensile modulus of the samples was approximately 9.29 GPa. Nevertheless, the tensile strength was included in our analysis, because in the case of electrospun interlayer an increase of tensile strength was observed (about 3%), which is attributed to the extra load-transfer mechanisms that nanofibers introduce to the CFRP composite (see S3 Fig and S3 Table in the supporting information S1 File). The epoxy foamed interlayer, especially with CNTs was observed to cause an increase in the values of energy release rate with large scattering. This phenomenon is explained by the fact that foaming interlayer is "non-homogeneous" both in terms of voids distribution within the foaming interlayer and in terms of voids size (see S4 Fig in the supporting information S1 File). During fracture toughness mode II experiments, the crack starts propagating within the interlayer in different manners: (1) In certain cases, the crack can reach to a CNT reinforced "small" void which can be a crack inhibitor requiring more energy for the crack to propagate and resulting in high energy release rates. (2) In other cases, the crack can propagate between the voids without reaching a void or reach to "large" voids, which facilitates the crack propagation resulting in lower energy release rates.

Feature permutation analysis indicated that the CNT volume fraction had minimal impact on the flexural strength compared to other independent variables. The feature permutation

importance measures the decrease in the model score when one feature is randomly shuffled, which breaks the relationship between this feature and the target value. In that case, a drop in the model score indicates a dependence of the model on that feature. Our feature permutation analysis also showed that the flexural modulus is only a function of the pressure, the glass transition temperature, and the interlayer volume fraction. This can be verified from Eq. (16), which assigns a relatively small coefficient to the volume fraction of CNTs, indicating the possibility of neglecting this term from the equation. Finally, additional independent variables that resemble the Type must be included in the model to accurately predict the mode II energy-release rate (Table 4 and Fig 8).

This study demonstrated that the glass transition temperature must be carefully controlled to develop CFRP composites with enhanced flexural strength. Koirala et al. [66] demonstrated that enhancing the interlaminar performance of CFRP by integrating ultra-thin sheets of CNTs with a weighted percentage of less than 0.05 improved the flexural strength of the CFRP by 49%. They also demonstrated that improving the interlayer volume fraction improves flexural strength. However, we reveal that the glass transition temperature is more important. To achieve an enhanced flexural modulus, the pressure and interlayer volume fraction should also be carefully controlled in addition to the glass transition temperature. To achieve an enhanced mode II energy-release rate, the interlayer volume fraction plays the most important role which agrees well with other studies in the literature [17,67].

The predicted results obtained by the three data-driven models were very consistent, particularly for flexural strength and flexural modulus (Table 3), which validates the robustness of the proposed predictive models. Changing the number of estimators of the random forest model did not have a significant effect on the output. The three ML models were selected to deal with limited amount of data so that they allow for a margin of error represented by ($\frac{\lambda}{2}\beta^T\beta^2$ term in the ridge regression, the threshold value $\varepsilon$ in support vector regression, and ensemble learning of 100 decision trees in the random forest model). Removing the Type variable improved the prediction accuracy of flexural strength and modulus in ridge regression and support vector machine and decreased the accuracy for the random forest models. However, as alluded earlier, the changes were slight for the flexural strength and modulus compared with the mode II energy release rate. In addition, our results showed that for certain properties such as flexural strength and flexural modulus, both linear (ridge regression) and nonlinear (RF) models provided comparable results (see Table 3). Table 3 also shows that for mode II energy release rate, the RF model yielded the highest $R^2$ value. Being an Ensemble model, the RF generally reduces the risk of overfitting. The RF model was also reported by [68] to demonstrate best results in predicting the tensile strength and elastic modulus for continuous fiber-reinforced polymer matrix composites.

The data-driven models developed inthis study effectively contribute to the prediction of the flexural strength, flexural modulus, and mode II energy release rate of CFRP, eliminating the need for costly experimental procedures. The corresponding $R^2$ values indicate acceptable predictive accuracy for all three properties, demonstrating the model's reliability and applicability. Future work could focus on (1) scalability and increasing the amount of data to evaluate the generalizability of the models. Recent work proposing specialized data augmentation techniques for composites could be helpful for this direction [69]; (2) including other parameters as independent variables such as fractal dimension [63] to investigate their effect on the prediction of the mode II energy-release rate; and (3) studying the effect of CNTs, interlayer foaming, and electrospinning of fibers at higher manufacturing pressures.

## Conclusions

Data-driven models were developed for CFRP composites, including those with feather-weight structures. The models considered - ridge regression, random forest, and support vector regression - effectively modeled small data sets with a reasonable margin of error. These models accurately predicted the flexural strength ($R^2$ = 0.966) and flexural modulus ($R^2$ = 0.871) based on the volume fraction of carbon nanotubes, interlayer volume fraction, glass transition temperature, and manufacturing pressure. Our findings highlight the glass transition temperature as the most critical factor in improving flexural strength, followed by manufacturing pressure, interlayer volume fraction, and CNT volume fraction. A similar trend was observed for the flexural modulus, with the interlayer volume fraction being more significant than manufacturing pressure. For mode II energy-release rate, the interlayer volume fraction was the most important variable, followed by the Type, CNT volume fraction, and glass transition temperature. The inclusion of additional variables, such as the dummy variable Type, was necessary to accurately predict the mode II energy release rate ($R^2$ = 0.903), suggesting that Type may encapsulate intrinsic properties such as fractal dimension or void patters in the interlayer.

This study successfully addressed the research questions posed at the beginning. It demonstrated that machine learning models can predict the mechanical properties of CFRP composites with high accuracy using the selected variables. The study identified the most significant independent variables influencing these properties, particularly the glass transition temperature and the interlayer volume fraction. Including multiple variables in the models significantly improved predictive accuracy compared to models that focus on a single or limited factor.

However, the current study was bound with some limitations: (i) the unequal sample sizes of some groups due to exclusions after failed manufacturing or mechanical testing; (ii) the inaccessibility of the data to deduce the effect of the manufacturing pressure on the tensile strength and mode II energy release rate. The first limitation was mitigated by validating that the average values of the mechanical properties agree with the values reported from our previous work. The second limitation does not affect our conclusion on the tensile strength because we did not predict it successfully in this study. The proposed predictive model for the mode II energy release rate should be carefully treated because it did not include data covering different manufacturing pressures.

The current study underscores the significant role of generalizable data-driven models in predicting the mechanical properties of CFRP composites, where traditional physics-based models fail to capture highly non-linear relationships among the variables. Our findings pave the way for more efficient design and optimization of advanced composite materials, reducing the reliance on extensive experimental testing. Future work should focus on scaling the dataset and incorporating additional parameters to further enhance the robustness and applicability of predictive models.

## Supporting information

**S1 File.  Multilinear regression validation.**  This file contains supplementary results on multilinear regression analysis, including statistical assumption checks, collinearity analysis, and a correlation matrix. **S1 Fig.** Validity of the regression's assumptions. **S2 Fig.** Durbin-Watson test. **S3 Fig.** Tensile stress-strain curves for 5 types of CFRP composites. **S4 Fig.** Scanning electron microscopy (SEM) of CFRP samples with epoxy foamed interlayer. **S1 Table.** Summary of the p-values corresponding to each dependent variable where $x_1$: carbon nanotube volume fraction, $x_2$: interlayer volume fraction, $x_3$: glass transition temperature and $x_4$: manufacturing pressure. **S2 Table.** Correlation matrix of independent variables where $x_1$: carbon nanotube

volume fraction, $x_2$: interlayer volume fraction, $x_3$: glass transition temperature, and $x_4$: manufacturing pressure. **S3 Table.** Tensile modulus for five types of CFRP composites.
(DOCX)

## Author contributions

**Conceptualization:** Haris Doumanidis, Maher Maalouf.

**Data curation:** Vassilis Drakonakis.

**Formal analysis:** Ammar Alsheghri, Amna Alhammadi, Imad Barsoum.

**Funding acquisition:** Vassilis Drakonakis.

**Investigation:** Ammar Alsheghri, Amna Alhammadi, Vassilis Drakonakis.

**Methodology:** Haris Doumanidis, Imad Barsoum, Maher Maalouf.

**Resources:** Vassilis Drakonakis.

**Software:** Ammar Alsheghri.

**Supervision:** Imad Barsoum, Maher Maalouf.

**Validation:** Ammar Alsheghri.

**Visualization:** Ammar Alsheghri.

**Writing – original draft:** Ammar Alsheghri, Amna Alhammadi.

**Writing – review & editing:** Vassilis Drakonakis, Haris Doumanidis, Imad Barsoum, Maher Maalouf.

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
