## [Decision Letter · Decision Letter 0]

18 Dec 2024

PONE-D-24-41071Predicting Mechanical Properties of CFRP Composites Using Data-Driven Models with Comparative AnalysisPLOS ONE

Dear Dr. Maalouf,

Thank you for submitting your manuscript to PLOS ONE. After careful consideration, we feel that it has merit but does not fully meet PLOS ONE’s publication criteria as it currently stands. Therefore, we invite you to submit a revised version of the manuscript that addresses the points raised during the review process.

**ACADEMIC EDITOR: **

Please read and address all comments provided by the reviewers. Special attention should be given to the literature review and the details of the experiments. Note that you are not required to cite all the papers requested by the reviewers. Only if you find them relevant should you add them to your manuscript.

Reason for the changes<svg aria-hidden="true" class="transition-transform ms-auto" fill="currentColor" focusable="false" height="8" style="display: inline-block; user-select: none; vertical-align: text-bottom; overflow: visible;" viewbox="0 0 12 12" width="8"><path d="M6 8.825c-.2 0-.4-.1-.5-.2l-3.3-3.3c-.3-.3-.3-.8 0-1.1.3-.3.8-.3 1.1 0l2.7 2.7 2.7-2.7c.3-.3.8-.3 1.1 0 .3.3.3.8 0 1.1l-3.2 3.2c-.2.2-.4.3-.6.3Z"></path></svg> 

We look forward to receiving your revised manuscript.

Kind regards,

Mohammadreza Vafaei, Ph.D.

Academic Editor

PLOS ONE

Journal Requirements:

3. We note you have included a table to which you do not refer in the text of your manuscript. Please ensure that you refer to Table 1 in your text; if accepted, production will need this reference to link the reader to the Table.

Reviewers' comments:

Reviewer's Responses to Questions

**Comments to the Author**

1. Is the manuscript technically sound, and do the data support the conclusions?

Reviewer #1: Yes

Reviewer #2: Partly

2. Has the statistical analysis been performed appropriately and rigorously? 

Reviewer #1: Yes

Reviewer #2: N/A

3. Have the authors made all data underlying the findings in their manuscript fully available?

Reviewer #1: Yes

Reviewer #2: Yes

4. Is the manuscript presented in an intelligible fashion and written in standard English?

Reviewer #1: Yes

Reviewer #2: Yes

5. Review Comments to the Author

Reviewer #1: Following points should be considered:

• Please mention the size of the dataset used in training the models in the Abstract.

• From Section 1. Introduction: "In polymerreinforced composites, the polymer matrix controls thermal and chemical properties, while the reinforcing material determines the mechanical properties" → Please provide some examples of these "thermal and chemical properties" and "mechanical properties"

• From Section 1. Introduction: "replacing metal-based materials in low-temperature applications due to these properties" → Please explain what these "low-temperature applications" are.

• From Introduction: "More recently, machine learning (ML) has become widely spread in predicting the mechanical properties of composite materials, significantly saving experimental and computational costs[11,12]" → At this point the authors should provide a much more comprehensive list of relevant publications since ML methods have been extensively used in the predictive modeling of composite materials. I recommend adding the following recent publications:

https://doi.org/10.1038/s41598-021-85963-3

https://doi.org/10.3390/ma16134578

https://doi.org/10.1002/pc.27969

https://doi.org/10.3390/biomimetics9090544

https://doi.org/10.1016/j.compositesb.2023.111132

• From Introduction: "Pathan et al. [17] found that the time required to predict

the mechanical properties using a machine learning model is 3600 times faster than using a finite element model." → This statement should be justified and further details should be provided about the measurement procedure. Although predicting the behavior of a model using trained models can be a fast process, the ML process also includes the data collection, preprocessing and model validation stages. Looking at the issue from this perspective, I don't think ML process is 3600 times faster than finite element analysis. Although finite element model building can be a time consuming process, once a model has been validated, running structural analysis simulations is a very fast process as well. Therefore, I recommend including more detailed information about in what way the ML process is faster than finite element analysis.

• I recommend, if available, including the actual pictures of the experimental setup in addition to drawing in Figure 1 and Figure 3. Also, Figure 1 appears 2 times. Please remove one of them.

• In addition to hardware used in training the models, please also mention the software platform used such as Scikit learn, Tensorflow etc.

• In the Discussion section the authors are proposing predictive formulas which should be done earlier in the text. I suggest moving that part into Results section or into a separate section before Discussion.

Reviewer #2: Alsheghri et al. applied ML models to predict polymer matrix-carbon reinforced composites (PMCRC). The input variables were manufacturing method and volume fiber content, whereas the output variables were 3-point flexural strength and modulus and the mode II energy release rate. Prediction metrics was improved after including manufacturing type. However, experimental details are missing; the literature review left out some recent developments / currently accepted methods and needs more critique of the current state. Some other issues need to be resolved before it can be considered for publication.

Experimental details are missing: load cell, extensometer/deflection gauge model, mouth opening gauge/crack length measurement device, acquisition and processing software.

Fig 1, Fig 2 are repeated. It probably happened when you duplicate the marker in MS-Word.

Fig 8b ... Feature premutatio ....

Are the bending samples manufactured/tested according to a standard? Please state that explicitly.

Major

The literature revision seems to focus on speed prediction rather than prediction accuracy. Furthermore, some accepted models, such as the Voight rule or micromechanics, are used for the same tasks that are not mentioned.

The authors performed tensile tests but omitted the tensile elastic modulus from calculation and prediction. Such comparison between flexural and tensile elastic modulus is always helpful, with the significance of each one depending on the PMCRC´s specific application.

The authors left recent advances in mechanical properties prediction using ML (Prada 2024), who used some of the models tested here for similar tasks. They use non-linear algorithms, which proved to be a better solution when predicting strength and elasticity (Prada 2024). Moreover, Random Forest and NN, are known for overfitting data.

The energy release rate is crucial for fail-safe designs. It makes sense that the manufacturing method impacts it, as the matrix-fiber bond might be enhanced at higher pressures. The predictions shown in Figs 7 and 8 seem very tight below 1000 kJ/m^2 but scattered above that value. Without experimental details on how it was calculated, one cannot venture to guess the origin of the high dispersion at higher rates. I urge the authors to discuss such behavior.

***

Prada-Parra, D. Supervised Machine Learning Models for Mechanical Properties Prediction in Additively Manufactured Composites. Appl. Sci. 2024, 14, 7009. https://doi.org/10.3390/app14167009

6. PLOS authors have the option to publish the peer review history of their article (what does this mean? ). If published, this will include your full peer review and any attached files.

**Do you want your identity to be public for this peer review?** For information about this choice, including consent withdrawal, please see our Privacy Policy .

Reviewer #1: No

Reviewer #2: No

---

## [Author Response · Author response to Decision Letter 1]

23 Jan 2025

We thank the editor for their feedback. The following items are being submitted with the revised manuscript:

• A rebuttal letter that responds to each point raised by the academic editor and reviewer(s).

• A marked-up copy of the manuscript that highlights changes made to the original version. You should upload this as a separate file labeled 'Revised Manuscript with Track Changes'.

Comments

Response

- The references style was updated to the Vancouver style as recommended by the journal.

- Line numbering has been incorporated.

- Title and author affiliations formatted according to the style.

- “Figure” replaced with “Fig”

Response

Our data has been published and we are happy to provide the code upon request.

The following section has been added:

Data availability

The data used in this manuscript is publicly available at Mendeley Data [69] and python codes will be made available upon request.

3. We note you have included a table to which you do not refer in the text of your manuscript. Please ensure that you refer to Table 1 in your text; if accepted, production will need this reference to link the reader to the Table.

Response

This has been fixed in the revised manuscript as:

Table 1 lists the nomenclature for all the abbreviations used in this manuscript.

Response

- Supplementary materials document has been renamed to “Supporting information”.

- Captions were included for the supporting information in the revised version of the manuscript.

Responses to reviewers:

We thank the reviewers for the excellent feedback which will improve the quality of our paper. We here try our best to respond to the comments of the reviewer one-by-one. We also made highlighted modifications inside the manuscript, to directly address the comments of the reviewers.

Reviewer #1: Following points should be considered

• Please mention the size of the dataset used in training the models in the Abstract

The sample size has now been included in the abstract of the revised version.

• From Section 1. Introduction: "In polymer reinforced composites, the polymer matrix controls thermal and chemical properties, while the reinforcing material determines the mechanical properties" → Please provide some examples of these "thermal and chemical properties" and "mechanical properties"

Examples have been added in the revised version as recommended by the reviewer:

In general, for polymer-reinforced composites, the polymer matrix controls thermal and chemical properties such as glass transition temperature, corrosion resistance, and chemical stability [2,3], while the reinforcing material controls mechanical properties such as tensile strength and impact resistance [4–7].

• From Section 1. Introduction: "replacing metal-based materials in low-temperature applications due to these properties" → Please explain what these "low-temperature applications" are.

Explanations have been added in the revised version as suggested by the reviewer:

“Due to these properties, carbon fiber-reinforced polymer (CFRP) composites have been crucial in replacing metal-based materials in low-temperature applications such as thermal insulators [9], electronic packaging [10], biomedical implants [11], and cryogenic systems [12].”

• From Introduction: "More recently, machine learning (ML) has become widely spread in predicting the mechanical properties of composite materials, significantly saving experimental and computational costs[11,12]" → At this point the authors should provide a much more comprehensive list of relevant publications since ML methods have been extensively used in the predictive modeling of composite materials. I recommend adding the following recent publications:

https://doi.org/10.1038/s41598-021-85963-3

https://doi.org/10.3390/ma16134578

https://doi.org/10.1002/pc.27969

https://doi.org/10.3390/biomimetics9090544

https://doi.org/10.1016/j.compositesb.2023.111132

We agree with the reviewer. The suggested recent publications have been added to the revised version of the manuscript with additional text:

“More recently, machine learning (ML) has become widely spread in predicting the mechanical properties of composite materials, significantly saving experimental and computational costs [6,7,20–24]. The application of ML also provided efficient solutions for the traditional difficulty associated with carrying out computational mechanics simulations involving multiscale and nonlinear behavior common in composites [6,7,22–24].”

• From Introduction: "Pathan et al. [17] found that the time required to predict the mechanical properties using a machine learning model is 3600 times faster than using a finite element model." → This statement should be justified and further details should be provided about the measurement procedure. Although predicting the behavior of a model using trained models can be a fast process, the ML process also includes the data collection, preprocessing and model validation stages. Looking at the issue from this perspective, I don't think ML process is 3600 times faster than finite element analysis. Although finite element model building can be a time consuming process, once a model has been validated, running structural analysis simulations is a very fast process as well. Therefore, I recommend including more detailed information about in what way the ML process is faster than finite element analysis.

The number reported “i.e. 3600” here refers only to the prediction stage after building and validating the FE and ML models. We have added more explanation to the revised version of the manuscript as recommended by the reviewer.

“Pathan et al. [29] found that the time required to predict five mechanical properties using a machine learning model was about 0.5 seconds, which was 3600 times faster than using a finite element model that took about 30 minutes to make the same predictions. This excludes the time required to build and validate the FE model or the time required for data collection, preprocessing, training and validation of the ML model.”

• I recommend, if available, including the actual pictures of the experimental setup in addition to drawing in Figure 1 and Figure 3. Also, Figure 1 appears 2 times. Please remove one of them

Pictures have been added as recommended. Please note that we are submitting the revised version of the article without the figures and attaching the figures separately.

• In addition to hardware used in training the models, please also mention the software platform used such as Scikit learn, Tensorflow etc.

Software packages have been added in the revised version of the manuscript:

“Pandas, scipy, and scikit-learn python packages were used to process the data and build the ML models.”

• In the Discussion section the authors are proposing predictive formulas which should be done earlier in the text. I suggest moving that part into Results section or into a separate section before Discussion.

We move the predictive formulas with the corresponding text to end of the Results section as suggested by the reviewer.

---

## [Decision Letter · Decision Letter 1]

2 Feb 2025

PONE-D-24-41071R1Predicting Mechanical Properties of CFRP Composites Using Data-Driven Models with Comparative AnalysisPLOS ONE

Dear Dr. Maalouf,

Thank you for submitting your manuscript to PLOS ONE. After careful consideration, we feel that it has merit but does not fully meet PLOS ONE’s publication criteria as it currently stands. Therefore, we invite you to submit a revised version of the manuscript that addresses the points raised during the review process.

**ACADEMIC EDITOR: **

Please check the English language and consistency of the symbols used in the manuscript.

We look forward to receiving your revised manuscript.

Kind regards,

Mohammadreza Vafaei, Ph.D.

Academic Editor

PLOS ONE

Journal Requirements:

Additional Editor Comments:

Please check the English language and consistency of the symbols used in the manuscript.

Reviewers' comments:

Reviewer's Responses to Questions

**Comments to the Author**

1. If the authors have adequately addressed your comments raised in a previous round of review and you feel that this manuscript is now acceptable for publication, you may indicate that here to bypass the “Comments to the Author” section, enter your conflict of interest statement in the “Confidential to Editor” section, and submit your "Accept" recommendation.

Reviewer #1: All comments have been addressed

Reviewer #2: All comments have been addressed

2. Is the manuscript technically sound, and do the data support the conclusions?

Reviewer #1: Yes

Reviewer #2: Yes

3. Has the statistical analysis been performed appropriately and rigorously? 

Reviewer #1: Yes

Reviewer #2: N/A

4. Have the authors made all data underlying the findings in their manuscript fully available?

Reviewer #1: (No Response)

Reviewer #2: No

5. Is the manuscript presented in an intelligible fashion and written in standard English?

Reviewer #1: Yes

Reviewer #2: Yes

6. Review Comments to the Author

Reviewer #1: (No Response)

Reviewer #2: The authors submitted a revised version. They addressed all observations and concerns raised in the first draft. I recommend it for publication upon checking English spelling, such as:

L396. “mode II release rate” should not go in capital letters. Furthermore, please make consistent its capitalization.

Eq (15). there is a ++

L564 gneralizable

7. PLOS authors have the option to publish the peer review history of their article (what does this mean? ). If published, this will include your full peer review and any attached files.

**Do you want your identity to be public for this peer review?** For information about this choice, including consent withdrawal, please see our Privacy Policy .

Reviewer #1: No

Reviewer #2: No

---

## [Author Response · Author response to Decision Letter 2]

3 Feb 2025

Reviewer #1: (No Response)

Reviewer #2: The authors submitted a revised version. They addressed all observations and concerns raised in the first draft. I recommend it for publication upon checking English spelling, such as:

L396. “mode II release rate” should not go in capital letters. Furthermore, please make consistent its capitalization.

Eq (15). there is a ++

L564 generalizable

We thank the reviewer for noticing these details. We have corrected them all and fixed some other minor English issues in the revised manuscript.

---

## [Decision Letter · Decision Letter 2]

9 Feb 2025

Predicting Mechanical Properties of CFRP Composites Using Data-Driven Models with Comparative Analysis

PONE-D-24-41071R2

Dear Dr. Maher Maalouf

We’re pleased to inform you that your manuscript has been judged scientifically suitable for publication and will be formally accepted for publication once it meets all outstanding technical requirements.

Kind regards,

Mohammadreza Vafaei, Ph.D.

Academic Editor

PLOS ONE

Additional Editor Comments (optional):

Reviewers' comments:

Reviewer's Responses to Questions

**Comments to the Author**

1. If the authors have adequately addressed your comments raised in a previous round of review and you feel that this manuscript is now acceptable for publication, you may indicate that here to bypass the “Comments to the Author” section, enter your conflict of interest statement in the “Confidential to Editor” section, and submit your "Accept" recommendation.

Reviewer #2: All comments have been addressed

2. Is the manuscript technically sound, and do the data support the conclusions?

Reviewer #2: Yes

3. Has the statistical analysis been performed appropriately and rigorously? 

Reviewer #2: Yes

4. Have the authors made all data underlying the findings in their manuscript fully available?

Reviewer #2: No

5. Is the manuscript presented in an intelligible fashion and written in standard English?

Reviewer #2: Yes

6. Review Comments to the Author

Reviewer #2: The authors submitted a revised version. They addressed all observations and concerns raised in the first draft. I recommend it for publication.

7. PLOS authors have the option to publish the peer review history of their article (what does this mean? ). If published, this will include your full peer review and any attached files.

**Do you want your identity to be public for this peer review?** For information about this choice, including consent withdrawal, please see our Privacy Policy .

Reviewer #2: No

---

## [Editor Report · Acceptance letter]

PONE-D-24-41071R2

PLOS ONE

Dear Dr. Maalouf,

I'm pleased to inform you that your manuscript has been deemed suitable for publication in PLOS ONE. Congratulations! Your manuscript is now being handed over to our production team.

Kind regards,

on behalf of

Dr. Mohammadreza Vafaei

Academic Editor

PLOS ONE